# Antioxidant Activity in Supramolecular Carotenoid Complexes Favored by Nonpolar Environment and Disfavored by Hydrogen Bonding

**DOI:** 10.3390/antiox9070625

**Published:** 2020-07-16

**Authors:** Yunlong Gao, A. Ligia Focsan, Lowell D. Kispert

**Affiliations:** 1College of Sciences, Nanjing Agricultural University, Nanjing 210095, China; 2Department of Chemistry, Valdosta State University, Valdosta, GA 31698, USA; 3Department of Chemistry & Biochemistry, The University of Alabama, Tuscaloosa, AL 35487, USA; lkispert@ua.edu

**Keywords:** carotenoids, oxidation potentials, antioxidant activities, photostability, supramolecular complexes, β-glycyrrhizic acid, liposomes, MCM-41, TiO_2_, polarity of environment, hydrogen bond, anchoring mode

## Abstract

Carotenoids are well-known antioxidants. They have the ability to quench singlet oxygen and scavenge toxic free radicals preventing or reducing damage to living cells. We have found that carotenoids exhibit scavenging ability towards free radicals that increases nearly exponentially with increasing the carotenoid oxidation potential. With the oxidation potential being an important parameter in predicting antioxidant activity, we focus here on the different factors affecting it. This paper examines how the chain length and donor/acceptor substituents of carotenoids affect their oxidation potentials but, most importantly, presents the recent progress on the effect of polarity of the environment and orientation of the carotenoids on the oxidation potential in supramolecular complexes. The oxidation potential of a carotenoid in a nonpolar environment was found to be higher than in a polar environment. Moreover, in order to increase the photostability of the carotenoids in supramolecular complexes, a nonpolar environment is desired and the formation of hydrogen bonds should be avoided.

## 1. Introduction

Carotenoids are a group of compounds widely existing in nature. They are based on a C40-tetraterpenoid skeleton and are usually classified into two main groups: hydrocarbon carotenoids, known as carotenes (e.g., β-carotene and lycopene), and carotenoids containing oxygen, known as xanthophylls (e.g., canthaxanthin and lutein). The majority of about 700 characterized carotenoids are natural compounds synthesized by plants and microorganisms that confer the yellow, orange and red colors. Examples of synthetic carotenoids with other functional groups added in the lab are 7′-apo-7′,7′-dicyano-β-carotene containing two cyano groups, and 8′-apo-β-caroten-8′-oic acid containing one carboxyl group.

About 20 carotenoids have been detected in the human blood stream and tissues [1]. Carotenoids are health promoters. Due to their long conjugated chains, carotenoids are highly reactive and efficient scavengers of free radicals [2,3,4]. In this process, the carotenoids react chemically with the free radicals and the system of conjugated double bonds is directly destroyed. Diseases such as cancers, cerebral thrombosis and infarction are partly a result of the action of free radicals and other reactive oxygen species (ROS) [5]. A small percent of the adsorbed oxygen in the lungs is used to make harmful ROS, such as hydrogen peroxide (H_2_O_2_) and the superoxide radical anion (O_2_^•−^) [5]. When these ROS react with transition metals like iron and copper in the human body, very reactive ROS such as hydroxyl ^•^OH radicals are produced. These radicals are very harmful to cells in the human body [5,6].

Carotenoids have become very popular nutritional supplements in the food and pharmaceutical industries. Carotenoids are also highly appreciated in the cosmetic industry owing to their bright colors, nutrition and absorption of UV light. For example, carotenoid bixin is used in cosmetic compositions in order to protect the human epidermis against UV radiation [7]. In the 1938 Federal Food, Drug and Cosmetic Act and in the 1960 Color Additive Amendments, bixin was approved as a suitable colorant for use in food, drugs and cosmetics [8]. β-Carotene is used in the formulation of bath products, aftershave lotions, makeup, cleansing products, hair shampoos, conditioners, skin care products and suntan products. It imparts an orange color to the cosmetic products, and also reduces flaking and restores suppleness to enhance the appearance of dry or damaged skin [9].

Carotenoids are electron donors forming carotenoid radical cations by oxidation, and reducing an oxidizer’s higher oxidation state to a lower oxidation state. The values of redox potentials, either expressed as the reduction potential of the radical cation or as the oxidation potential of the neutral carotenoid, can be important indicators of their antioxidant efficacy. Truscott’s group [10] has determined the one-electron reduction potentials of the radical cations of different carotenoids in aqueous micellar to be in the range of 980–1060 mV, with β-carotene radical cation being 1060 ± 25 mV. The one-electron reduction potential of the radical cation of β-carotene with the same value was also measured for the first time in a model biological aqueous environment by the same group [11]. Previously, Mairanovski et al. [12] measured two-electron reduction potentials of carotenoids mainly in halogenated organic solvents. In our lab, we determined the first oxidation potentials of neutral carotenoids using cyclic voltammetry in methylene chloride measured against calomel electrode. Others have confirmed similar values for the first oxidation potentials of carotenoids [13] measured under similar conditions, or trends in antioxidant activity by either experiments or calculations [14,15,16,17,18]. The literature in the past decade reporting redox potentials of carotenoids remains scarce.

It was determined that carotenoids exhibit scavenging ability towards the free radicals that increases nearly exponentially with increasing the carotenoid oxidation potential [19,20]. We have recently found that *cis*-bixin exhibits the highest carotenoid oxidation potential (0.94 V vs SCE) to date [21]. Its scavenging ability towards ROS such as ^•^OH, ^•^OOH and O_2_^•−^ was 17 times higher than that of astaxanthin with an oxidation potential of 0.768 V, and 69 times higher than that of β−carotene with an oxidation potential of 0.63 V [21]. Carotenoids are also used to prolong the shelf life of pharmaceuticals based on their ability to scavenge free radicals [4,19,20]. Although carotenoids show abilities of quenching singlet oxygen and scavenging free radicals to prevent oxidation, under certain conditions, the antioxidant effect of carotenoids is weakened and even changed to the prooxidant effect. Early epidemiological studies have shown that supplementing smokers or asbestos patients with large amounts of β-carotene has no protective effect and increases the risk of lung cancer [22]. A certain level of lycopene was shown to promote oxidation, too [23]. The antioxidant and prooxidant effects of carotenoids are related to a multitude of factors such as the concentration of carotenoids, the molecular structure, the action sites, oxygen pressure, the interaction of carotenoids with other dietary antioxidants, and the methods used to induce oxidative stress [24]. We would like to add here the importance of the oxidation potential of the carotenoid. In our studies, it was shown that because of its low oxidation potential, β−carotene exhibits a prooxidant behavior, the radical cation of β−carotene being formed in reaction with Fe^3+^ [20]. In a Fenton reaction, carotenoids with low oxidation potentials like β−carotene reduce Fe^3+^ to Fe^2+^ to form carotenoid radical cations. The regeneration of Fe^2+^ restarts the Fenton chemistry, which causes an overall increase in the radical production. In the Fenton reaction, carotenoids with high oxidation potentials, like astaxanthin, exhibit an antioxidant behavior—they scavenge free radicals via proton abstraction rather than reducing Fe^3+^, thus leading to a decrease in the total radical yield. Additionally, it was shown that aggregation of xanthophylls significantly reduces the antioxidant activity. For example, zeaxanthin aggregates show prooxidant effect instead of antioxidant activity [25] in the presence of Fe^2+^ ions and hydrogen peroxide in a Fenton reaction.

Some supramolecular carotenoid complexes and the effect of complexation on the antioxidant activity are described in our recent review article [26]. In order to increase antioxidant activity and bioavailability of carotenoids as well as stability towards irradiation and ROS, carotenoids can be incorporated into host molecules such as cyclodextrin (CD) [27], arabinogalactan (AG) [28] and β-glycyrrhizic acid (GA) [4,29,30] (Figure 1). Incorporation of carotenoids in host molecules enhances not only the water solubility but also the oxidation stability and photostability of carotenoids. The complexes of carotenoids with AG enhanced photostability by a factor of 10 in water solutions, and significantly decreased reactivity towards metal ions (Fe^3+^) and ROS in solution [28] by a factor of 20. The complexation with GA increased the oxidation potentials, which resulted in increased antioxidant activity of selected carotenoids. For example, the oxidation potentials of zeaxanthin and canthaxanthin in the presence of GA increased compared with those measured in dimethyl sulfoxide (DMSO) solution. The scavenging rate constant of ^•^OOH hydroperoxyl radicals by GA complex was much larger than the free carotenoid [4,25]. Other benefits of the incorporation include prevention of aggregation of xanthophyll carotenoids in aqueous solutions, and a remarkable increase in the quantum yield and the lifetime of the charge-separated states of the carotenoid radical cations [31]. Carotenoids can also be encapsulated into more traditional delivery systems like lipid-based nanocarriers [32,33]. The lipid-based nanocarriers include nanoemulsions [34,35,36], nanoliposomes [37,38,39], solid lipid nanoparticles [40,41,42], and nanostructured lipid carriers [43,44,45] (Figure 2). A study by Tan and coauthors [38] found that the encapsulation of carotenoids in liposomes enhanced the antioxidant activity in different antioxidant models. The enhancements are various for different carotenoids depending on the orientations of carotenoids in the models. Some non-traditional hosts like mesoporous molecular sieves MCM-41 and TiO_2_ discussed here provided us with information on electron transfer (ET) efficiency from the carotenoid to the host, and explained how hydrogen bonds (H-bonds) and anchoring modes affect the photostability of the carotenoids. The ET efficiency is enhanced when carotenoids act as H-bond donors, and reduced if carotenoids act as H-bond acceptors, which significantly affects the photostability of carotenoids in those hosts.

As mentioned above, the scavenging ability of carotenoids towards free radicals is strongly dependent on the oxidation potential [19,20]. Complexation with GA was shown to affect the oxidation potentials of the carotenoids and thus the antioxidant activity [4,25]. The encapsulation of carotenoids in other delivery systems also enhanced the antioxidant activity [38,39]. Therefore, the oxidation potentials of carotenoids are a critical factor that determines the antioxidant activity of carotenoids. Since the ability of scavenging free radicals and shelf lives of carotenoids in delivery systems are both related to the antioxidant activities of the carotenoids, the factors affecting the oxidation potentials of carotenoids need to be examined. This paper summarizes studies performed in our lab in the last two decades on the oxidation potentials of carotenoids and the different factors affecting them. It explores how the chain lengths and donor/acceptor substituents of carotenoids affect the oxidation potentials of carotenoids. Recent progress on how polarity of environment and orientation of carotenoids affect the oxidation potentials of carotenoids and subsequently the antioxidant activities, is also presented in the current paper. The goal of this paper is to provide guidance for designing novel supramolecular carotenoid complexes, by modifying hosts or searching for new delivery systems for carotenoids, as this is very important in the pharmaceutical, food and cosmetic industries.

## 2. Oxidation Potentials of Carotenoids as Functions of Conjugation Length and Electron Donor/Acceptor Substituents

Since the antioxidant activities and shelf lives of carotenoids are oxidation potential-dependent, and the antioxidant activities increase exponentially with the oxidation potential, the factors determining the oxidation potentials of carotenoids need to be investigated. In this section, the oxidation potentials of carotenoids as functions of conjugation length and electron donor/acceptor substituents are examined. The results will provide guidance for proper selection or synthesis of new carotenoids for application in foods, cosmetics and pharmaceuticals.

The first oxidation potentials (E^0^) of the selected carotenoids [21,46,47,48,49] shown in Figure 3 are different depending on the conjugation lengths and electron donor/acceptor substituents. The oxidation potentials were measured in CH_2_Cl_2_ by cyclic voltammetry (CV) with measurement error ±10 mV. The reference electrode used in the measurements was saturated calomel electrode (SCE). Calibration with ferrocene gave the potentials corrected to SCE [46]. In Figure 3, all the oxidation potentials related to SCE are listed from the highest value 940 mV for 9′-cis bixin to the lowest value 593 mV for lycopene. The conjugation length can be measured by the number of double bonds (NDB) in the conjugated chain. The higher the NDB, the longer the conjugation length is. From Figure 3, it can be determined that carotenoids with similar structures having shorter conjugation lengths have higher oxidation potentials. For example, canthaxanthin and rhodoxanthin have similar structures and both contain two carbonyl groups at each of the two cyclohexene rings, but the NDB of the former (13) is 1 less than the latter (14), which results in higher oxidation potential for canthaxanthin (775 vs. 741 mV). Another example, lutein and zeaxanthin, which are isomers, have different oxidation potentials: lutein with one less double bond in the conjugated system has a higher oxidation potential compared to that of zeaxanthin (651 vs. 616 mV). The oxidation potential of a molecule is related to the HOMO (high occupied molecular orbital) energy of the molecule, and the lower the HOMO energy is, the higher the oxidation potential. A study by Méndez-Hernández et al. [50] shows that there is a very strong linear correlation of DFT-calculated HOMO and LUMO (lowest unoccupied molecular orbital) energies (HLE) and redox potentials of 51 polycyclic aromatic hydrocarbons (PAHs). The strong correlation obtained from the HLE and redox potentials of PAHs was independent of whether the solvent model was included in the calculations. A carotenoid with shorter conjugation length has lower HOMO energy, and thus has higher oxidation potential.

The oxidation potentials are also higher for those carotenoids containing electron acceptor or withdrawing substituents. The more electron-accepting groups a carotenoid contains, the higher the oxidation potential for the carotenoid. For example, 9′-cis bixin owns the highest oxidation potential because it contains four oxygen atoms with strong electronegativity in the conjugated system, and also the conjugation length is short with NDB being 11. Echinenone contains one less electron-withdrawing carbonyl group than canthaxanthin, and thus the oxidation potential is much lower than that of canthaxanthin (676 vs. 775 mV), although the conjugation length is slightly shorter than that of canthaxanthin (the NDBs are 12 and 13, respectively). The allenyl group in fucoxanthin and the alkynyl groups in 15,15’-didehydro-β-carotene are also electron-withdrawing groups because the allenyl carbon and the alkynyl carbon are both sp hybridized. The sp hybridized orbital contains more s components than the sp^2^ hybridized orbital, and shows stronger electron-withdrawing ability. The higher oxidation potentials for fucoxanthin and 15,15’-didehydro-β-carotene (876 and 875 mV, respectively) are attributed to the groups, in addition to the shorter conjugation lengths (NBDs are 9 and 11, respectively). This point is more clear if we compare 15,15’-didehydro-β-carotene with β-carotene. The conjugation lengths are the same, but 15,15’-didehydro-β-carotene contains two alkynyl carbons, resulting in much higher oxidation potential than β-carotene (875 vs. 634 mV). The high oxidation potential for fucoxanthin is also owing to the carbonyl group it contains. On the contrary, the oxidation potential of a carotenoid is reduced if it contains one or more electron-donating groups. For example, 7’-Apo-7’,7’-dimethyl-β-carotene contains two electron-donating methyl groups, and thus the oxidation potential is much lower than that of 7’-apo-7’,7’-dicyano-β-carotene (654 vs. 825 mV) containing two electron-withdrawing cyano groups, although the conjugation lengths of the two compounds are the same with the NDBs being 10.

## 3. Oxidation Potentials of Carotenoids in Polar and Nonpolar Environments

It is important to know how different environments affect the oxidation potentials of carotenoids and thus their antioxidant activities. The incorporation of carotenoids into hosts results in noncovalent binding like hydrophobic forces, van der Waals interactions or hydrogen bonds between the nonpolar carotenoid and the hosts [29]. Thus, the polarity of a host affects the stabilities of the neutral species and those of the radical cations, which in turn affects the oxidation potentials of carotenoids. In our most recent study [51], the polarities of the environments were simulated by adding the carotenoids in solvents with different polarities. By comparing the calculated oxidation potentials of carotenoids in different solvents like cyclohexane (C_6_H_12_), methylene chloride (CH_2_Cl_2_) and water (H_2_O) (see Table 1), the effect of polarity of different environments on the oxidation potential of the carotenoid was determined. The carotenoids used in the DFT calculations were retinol (I), 8’-apo-β-caroten-8’-ol (II), 4,4’-diapo-β-carotene (III) and β-carotene (IV) (see Table 1 and Figure 4). The structures of retinol and 8’-apo-β-caroten-8’-ol are unsymmetrical, and those of 4,4’-diapo-β-carotene and β-carotene are symmetrical. The conjugated chains of retinol and 4,4’-diapo-β-carotene are short, and those of 8’-apo-β-caroten-8’-ol and β-carotene are long. These carotenoids were chosen to determine whether the symmetry of the structures and the lengths of the conjugated chains play any roles. The solvents used in the calculations were C_6_H_12_, CH_2_Cl_2_ and H_2_O. Generally, the dielectric constant (ε) of a solvent provides a rough measure of the solvent’s polarity (the greater the dielectric constant, the greater the polarity) [52]. The ε of C_6_H_12_ is 2.02, indicating cyclohexane is a non-polar solvent, while the large ε of 78.36 indicate water is a very polar solvent. The ε of CH_2_Cl_2_ is 8.93, and its polarity is between those of C_6_H_12_ and H_2_O.

DFT calculations with the density functional M06-2X + D3 [53] and the C-PCM continuum solvation model [54] were shown to provide accurate predictions of the oxidation potentials of carotenoids in solvents with different polarities [51]. The relationships between the oxidation potentials versus ferrocene (E^0^_(Fc/Fc+)_) and the dielectric constants (ε) of the solvents for the carotenoids are revealed in Figure 4. The E^0^_(Fc/Fc+)_ values drop very significantly when the ε is increased from 2 to 9, however, they decrease slowly when ε is increased from 9 to 80, suggesting that the oxidation potentials are more sensitive to the dielectric constants in non-polar environments (i.e., the ε values are small). According to the calculations, the decrease in the oxidation potentials of the carotenoids with the increase in the polarity of a solvent is because the polar radical cations of the carotenoids are more stabilized than the neutral carotenoids in a polar solvent, resulting in more negative solvation-free energies compared with the neutral carotenoids [51]. The similar curves for the four carotenoids shown in Figure 4 indicate that these behaviors are independent of the symmetries and chain lengths of the carotenoids.

This new study [51] provides the quantitative data on the change of the oxidation potentials with different polarities of environments. The change from a very polar environment like H_2_O to a nonpolar environment like C_6_H_6_ is as large as 0.6 V. Since the scavenging ability of a carotenoid towards the free radicals increases nearly exponentially with increasing carotenoid oxidation potential, the increase of 0.03–0.05 V causes the scavenging rate constant to increase about 30 times [4,25]. Based on the current study, it is possible to estimate the oxidation potential of a carotenoid and thus the antioxidant activity if the polarity of the environment is known. The hosts for carotenoids mentioned above can be modified so that the polarities of the hosts are less polar. For example, the polar –OH groups in CD and AG can be replaced by less polar –OCH_3_ or –OCH_2_CH_3_ groups via esterification; the polar –COOH groups in GA can also be replaced by less polar –COOCH_3_ groups via esterification. The most significant part of the study is that because these results are independent of the symmetries and chain lengths of carotenoids, the conclusions are basically applicable to all drugs, nutrients or cosmetics containing a conjugated system.

Polyakov et al. have measured the scavenging rates of selected carotenoids towards ^•^OOH hydroperoxyl radical (equal to 0.64, 1.96, 3.22, 8.25, 12.4 and 24 M^−1^s^−1^ for β-carotene, canthaxanthin, 8′-apo-β-caroten-8′-al, 7,7′-diphenylcarotene, ethyl 8′-apo-β-caroten-8′-oate, and 7′-apo-7′,7′-dicyano-β-carotene, respectively) relative to a DMPO spin trap and they were found to be strongly potential-dependent [19,20]. It was then assumed that GA complexation can also affect the oxidation potentials of the carotenoids. To prove the hypothesis that GA complexation can affect the oxidation potentials of the carotenoids, the oxidation potentials of zeaxanthin and canthaxanthin in the presence of GA were measured by CV [4,25]. In both cases, an increase in the oxidation potentials by 0.03–0.05 V compared with those in polar DMSO (ε = 46.68) was observed. This is in accordance with our new findings [51], confirming that a nonpolar environment would increase the oxidation potential. It is known that in GA complexes, carotenoids interact with the nonpolar glycyrrhetinic acid residues (Figure 5a) which make the hydrophobic part of the dimer (Figure 5b), as shown in the optimized β-carotene-GA complex (Figure 5c). This results in higher oxidation potential than in polar DMSO. Another thing to be mentioned is that for carotenoids to play the antioxidant role, the hydrophilic ends of the carotenoids must be exposed to the surroundings because interaction between carotenoids and hydroperoxyl radicals occurs via hydrogen abstraction from the most acidic proton of carotenoids located at the hydrophilic ends [25,55]. GA forms a donut-like dimer in which the polyene chain of a carotenoid lies within the donut hole so that the hydrophilic ends are exposed to the surroundings (Figure 5c). For the cyclodextrin complexes, the terminal groups of carotenoids are blocked, which results in the inhibition of antioxidant activity [56].

The conclusions of this new study [51] also explain studies done by others [38]. The antioxidant activities of four carotenoids lutein, β-carotene, lycopene and canthaxanthin encapsulated in liposomes were investigated by Tan et al. [38]. The method to evaluate the antioxidant activities was 2,2-diphenyl-1-picrylhydrazyl (DPPH) radical scavenging. Compared with the carotenoids directly mixed with liposomes in DMSO solution, the carotenoids incorporated in liposomes exhibited higher DPPH radical-scavenging activity [38]. The enhancement of the antioxidant activities of the carotenoids can be attributed to the higher oxidation potentials of the carotenoids in the non-polar environment, because the carotenoids are within the non-polar hydrophobic bilayer of liposomes. Additionally, carotenoids exhibited various antioxidant activities in liposomes, ranging from the strongest to the weakest: lutein > β-carotene > lycopene > canthaxanthin. The authors attribute this to the position and orientation of the carotenoids in the bilayer, which were determined by an independent study [57]. In the current paper, a fundamental explanation is provided as follows based on the oxidation potentials, position and orientation of the carotenoids in the bilayer. The oxidation potentials vs. SCE in CH_2_Cl_2_ for canthaxanthin, lutein, β-carotene and lycopene are 775, 651, 634, 593 mV, respectively (see Figure 3). Moreover, according to Figure 4, the oxidation potentials of carotenoids increase in the non-polar environment, and the extent of increase should be similar because the curves are in parallel with each other. Therefore, the oxidation potentials of the four carotenoids in the bilayer of liposomes decrease in the following order: canthaxanthin > lutein > β-carotene > lycopene. Since the scavenging ability of a carotenoid towards free radicals increases nearly exponentially with increasing the carotenoid oxidation potential [19,20], it was expected that the antioxidant activities for the four carotenoids range from the strongest to the weakest: canthaxanthin > lutein > β-carotene > lycopene. The exception for canthaxanthin can be explained by its position and orientation in the bilayer. Canthaxanthin can adopt a horizontal orientation with respect to the plane of the lipid bilayer and link different polar lipid heads via hydrogen bonding, and it can also orientate vertically to the membrane plane through hydrogen bonding between their polar end groups and membrane’s polar region. The most acidic 3-H proton, which reacts with radicals, in canthaxanthin, is close to the surface of the lipid polar head when the hydrogen bonds form (see Figure 6), because canthaxanthin acts as a H-bond acceptor. Therefore, the access to the 3-H proton on the cyclohexene ring is blocked for the DPPH radical due to its large size. Lutein also adopts similar orientations to those of canthaxanthin via hydrogen bonding, however, lutein may act as a H-bond donor (see Figure 6) and the most acidic 4-H proton on the cyclohexene ring is far away from the surface of the lipid polar head. Thus, access to the 4-H proton is not blocked for the DPPH radical.

From the above discussion, it is demonstrated that polarity of the environment and position and orientation of carotenoids in the environment are extremely important in the determination of the antioxidant activities of the carotenoids, possibly more important than the value of the carotenoid oxidation potential. The molecular sizes of the free radicals may also affect the antioxidant activities. These factors must be considered in the design of supramolecular carotenoid complexes.

## 4. Carotenoids Imbedded in Mesoporous Molecular Sieves MCM-41

MCM-41 is a mesoporous silica containing a regular array of uniform cylindrical pores. The pore size ranges from 15 to 100 Å depending on the chain length of the template used in the synthesis [58]. MCM-41 and surface modified MCM-41 were found to be a good drug delivery system [59,60,61,62,63,64,65,66,67,68,69]. For example, Vallet-Regi et al. synthesized MCM-41 for charging Ibuprofen. Later, functionalization with aminopropyl group of MCM-41 was done in order to control its delivery rate. The release rate was different depending on the method for charging the drug but independent of the MCM-41 pore size [60,62]. In another study [69], MCM-41 coated with CaWO_4_:Ln (Ln = Eu^3+^, Dy^3+^, Sm^3+^, Er^3+^) phosphor layers designated as CaWO_4_:Ln@MCM-41 was used as delivery system for aspirin. The emission intensity of Eu^3+^ increased with the cumulative released aspirin and thus the release process could be monitored by the change in luminescence. 

Carotenoids can be imbedded in MCM-41 and metal ion-substituted MCM-41 [70,71,72,73]. Previous studies [70,71,74,75,76,77] have shown that such material provides a microenvironment appropriate for retarding back electron transfer (ET) and thus increases the lifetime of photo-produced radical ions. When carotenoids adsorb on the cylindrical pore surfaces of MCM-41 and metal ion-substituted MCM-41, chemical bonds, such as coordination bonds and hydrogen bonds, can form. Some carotenoids also physisorb on the surface. It is important to know how the formation of chemical bonds affects the efficiency of photo-induced ET from carotenoids to MCM-41. For example, β-carotene interacts with Cu^2+^ in the cylindrical pore of Cu-MCM-41 to form a complex, interaction detected by EPR measurements [71]. The formation of the complex favors light-driven ET from β-carotene to Cu^2+^ and also permits thermal back ET from Cu^+^ to the β-carotene radical cation [71]. However, when canthaxanthin is imbedded in Cu-MCM-41, it prefers to form one or two H-bonds with the silanol (–SiOH) groups on the MCM-41 surfaces rather than to form a complex with Cu^2+^. The formation of H-bonds is confirmed by DFT calculations, EPR measurements and calorimetric experiments [72]. DFT calculations show that the interaction energy (IE) due to the H-bonds is much lower than the IE between canthaxanthin and Cu^2+^, explaining why it prefers to form H-bonds [72]. The formation of the H-bond is due to the interaction between the oxygen atom on the cyclohexene ring of canthaxanthin and the proton of the -SiOH group.

It is necessary to investigate the photostability of carotenoids H-bonded or physisorbed on the surface of MCM-41 (see Figure 7) because it is related to the shelf life of the complex. Previous studies [72,73] showed that the formation of H-bonds affects the photo-induced ET efficiency (the net number of electrons transferred to the semiconductor by a certain amount of photons) of carotenoids adsorbed on mesoporous sieves on MCM-41. Carotenoids containing hydroxyl groups, such as retinol and 8’-apo-β-caroten-8’-al, and those containing ketone groups such as canthaxanthin, retinal and 8’-apo-β-caroten-8’-al, were adsorbed onto MCM-41 and surface-modified MCM-41. The H-bonding properties and charge separation efficiencies of those carotenoids were investigated by DFT calculations and EPR experiments [72,73], because the radical cations produced in the photo-irradiation are stable enough at low temperature to be detected by EPR techniques. Two types of H-bonds can be formed when a carotenoid containing an -OH group interacts with a -SiOH group on the MCM-41 surface, depending on whether the carotenoid is a H-bond donor or acceptor. Figure 7a shows the two types of H-bonds for retinol adsorbed on the surface of MCM-41. When the carotenoid is the H-bond acceptor, the formation of the H-bond decreases the energy of the LUMO of the carotenoid and stabilizes the neutral species more than the radical cation, and thus disfavors the photo-induced ET from the carotenoid to MCM-41. This effect is more pronounced if the oxygen atom of the carotenoid is completely conjugated with the conjugated chain. However, if the carotenoid is the H-bond donor, the formation of the H-bond increases the energy of the LUMO of the carotenoid and stabilizes the radical cation more than the neutral species and thus increases the charge separation efficiency significantly [73].

Although the formation of the H-bond gives lower charge separation efficiency when the carotenoid is the H-bond acceptor, the efficiency is still much higher than that for physisorbed carotenoid on the modified surface of MCM-41 (designated as R-MCM-41) with –SiOH groups replaced by –SiOCH_2_CH_3_ groups via esterification [78]. This was confirmed by comparison of the charge separation efficiencies of retinal and 8’-apo-β-caroten-8’-al imbedded in MCM-41 and R-MCM-41 [73]. Figure 7b shows the cartoons for 8’-apo-β-caroten-8’-al imbedded in the pores of MCM-41 and R-MCM-41. For 8’-apo-β-caroten-8’-al in MCM-41, a H-bond is formed with the carotenoid acting as the H-bond acceptor; for 8’-apo-β-caroten-8’-al in R-MCM-41, it physisorbs on the surface of R-MCM-41. The EPR spectra for the two samples (Figure 7c) [73] after 10 min photoirradiation shows that the intensity of the EPR signal for the carotenoid imbedded in MCM-41 is much stronger than that in R-MCM-41, which is barely detectable, suggesting that the ET efficiency in R-MCM-41 is much weaker than that in MCM-41. For 8’-apo-β-caroten-8’-al imbedded in R-MCM-41, there is no chemical bond formed between the carotenoid and R-MCM-41; the electron transfer is intermolecular and thus is slow because of the approximately exponential decrease in ET rate with increasing distance [79,80,81,82]. Another reason for the much lower charge separation efficiency is that the environment of R-MCM-41 is nonpolar due to the surface –SiOCH_2_CH_3_ groups, and thus it stabilizes the radical cation of the carotenoid less efficiently than the polar environment of MCM-41 with surface –SiOH groups, according to the recent study [51]. Therefore, the ET back transfer from the radical cation to R-MCM-41 is much faster than for MCM-41, resulting in much lower charge separation efficiency.

The above studies indicate that the types of H-bonds significantly affect the photoinduced ET efficiencies from carotenoids to hosts in supramolecular complexes. To increase the photostability of carotenoids, the formation of H-bonds should be avoided, especially when the carotenoids act as the H-bond donors. Nonpolar environments also favor photostability of carotenoids owing to the much lower photoinduced ET efficiency.

## 5. Carotenoids Anchored on TiO_2_

Although titania, TiO_2_, has been widely used as photocatalyst for environmental applications [83,84,85,86], TiO_2_ nanotube and mesoporous TiO_2_ were also used as drug delivery systems [87,88,89,90,91,92,93,94,95] because of their excellent properties and facile preparation process. For example, Song et al. [87] reported on amphiphilic TiO_2_ nanotubes used as biomolecular carriers in which the photocatalytic ability of TiO_2_ allowed a controlled release of a hydrophilic drug. Porous TiO_2_ nanoparticles modified with polyethylenimine (PEI) were used to encapsulate paclitaxel, a poorly soluble anti-cancer drug [88]. It was found that PEI on the surface could prevent premature drug release. The amount of the drug released could also be regulated by the UV-light radiation time. Yin et al. [90] synthesized mesoporous TiO_2_ nanoparticles for a near-infrared (NIR)-triggered drug delivery system while Xu and coauthors [92] used TiO_2_ nanotubes for a visible-light-triggered drug delivery system.

TiO_2_ is also utilized in the cosmetic industry. Nanosized TiO_2_ is used extensively in sunscreen cosmetics as an inorganic UV absorber that can allow an optically transparent film to be applied to human skin. TiO_2_ is known to exist in three crystal forms: anatase, brookite and rutile. The rutile phase is generally used as a component in sunscreen cosmetics because of its higher UV absorption. A surface coating can be added to nanosized TiO_2_ to enhance its UV absorption by a different light diffraction mechanism [96]. Leong et al. [97] prepared bifunctional alpha-bisabolol and phenylethyl resorcinol/TiO_2_ hybrid composites for anti-ageing and hyperpigmentation treatment. The functionalization of TiO_2_ particles with these skin-lightening materials produced a synergistic effect of combined antioxidant and UV filtering properties [97]. TiO_2_ nanoparticles were also used in anti-wrinkle day creams, lip balm protectors, toothpaste, etc. [98].

Since TiO_2_ is used in both pharmaceutical and cosmetic industries, the absorption and photostability of carotenoids adsorbed on TiO_2_ needed to be studied. Three main anchoring modes to the surface of TiO_2_ for carotenoids containing carboxylic acid groups adsorbed were investigated [99]. Figure 8 shows the optimized structures of carotenoid retinoic acid (RA) and the three complexes for RA [99] on the surface of a (TiO_2_)_16_ cluster [100]. The complexes were optimized by DFT calculations using a restricted B3LYP functional. The basis set 6–31G(d) was used for all atoms. For complex A, the carboxylic acid group loses a proton and forms two coordination bonds with two surface Ti atoms of TiO_2_. For complex B, the carboxylic acid group loses a proton and forms one coordination bond with one surface Ti atom of TiO_2_ and one hydrogen bond (H-bond) with a Ti-OH group on the surface of TiO_2_. The lost proton in A and B adsorbs on the surface of TiO_2_, so the whole system is neutral. The lost proton can be anywhere on the surface, but is likely near the negatively-charged oxygen atom. For complex C, the carboxylic acid group does not lose a proton and forms one coordination bond with one surface Ti atom of TiO_2_ and one H-bond, due to the interaction between the OH group of the carboxylic acid group and the bridging oxygen atom on the surface of TiO_2_. The calculated H-bonding properties (i.e., the O⋅⋅⋅H bond length and ∠O⋅⋅⋅H-O angle) and bond length of O-Ti for the coordination bonds are shown in the figure. The bond length of O-Ti for B (1.851 Å) is significantly shorter than those for A (1.966 Å and 2.026 Å) or for C (2.010 Å). The shortest O-Ti bond length for B was attributed to two reasons [99]. First, for B with one coordination bond and one H-bond formed, RA can orient relatively freely because the strength of the H-bond is much weaker than that of the coordination bond. By contrast, for A with two coordination bonds formed, RA cannot orient freely on the surface of TiO_2_. Second, RA in B loses a proton, and thus the charges on the oxygen atom forming the coordination bond are more negative than that for C, and the charges on the Ti atom are more positive for B due to the adsorbed proton on TiO_2_ (see Figure 8), which results in stronger electrostatic interaction between the oxygen atom and Ti^4+^ compared with C. It is important to know how each anchoring mode affects the efficiency of the photo-induced electron transfer (ET) from RA to TiO_2_, because the efficiency of ET is related to the photostability of the complexes.

Since the coupling of a dye excited states with semiconductor conduction band states increases as the distance between the dye’s LUMO and the conduction bands of the semiconductor decreases [101], the coupling is the strongest for B. The electronic coupling strength can be evaluated by the ratio of the electron density of the dye to that of TiO_2_. Shorter electron injection times were detected when the electron density contribution from the semiconductor was higher in the mixing state [102]. The study showed that for B, the contributions from TiO_2_ are much higher than in A and C. This was further confirmed by examining RA’s LUMO projected density of state (PDOS) in A, B and C, because the coupling between a dye and a semiconductor can broaden and split the dye LUMO [100,103]. In fact, according to the Newns−Anderson model [103,104], the injection rate can be estimated by simply considering the broadening of the dye’s LUMO PDOS, relative to the LUMO of the free dye [105,106].

The photo-induced ET efficiency is also strongly related to the maximum absorption wavelength (λ_max_) and the oscillator strength of RA, because these factors affect the excitation of RA. It was found that different anchoring modes for RA adsorbed on the surface of TiO_2_ significantly affected the bond lengths and thus the conjugation of RA. The degree of conjugation in a molecule can be evaluated by means of the bond length alternation (BLA) parameter [72], which is defined as the difference in total length between single bonds (C−C) and double bonds (C = C). The BLA parameter decreases as the degree of conjugation increases. The BLA parameter was calculated from C5 to C15. The calculated BLAs (see Figure 8) for RA, A, B and C are: 0.4879, 0.4231, 0.4662 and 0.4076 Å, respectively. The degree of conjugation in RA increases from B to A to C. The degree of conjugation affects the energies of RA’s HOMO and LUMO, and thus affects the UV/vis absorption of RA. TD–DFT calculations [99] showed that compared with the free RA, the maximum absorption wavelengths (λ_max_s) of A, B and C shift to the red because the degree of conjugation of RA anchored on TiO_2_ increases. The extents of shift are different for A, B and C, depending on the degrees of conjugation for the three complexes. This was further proved by the different colors of the complexes [99]. Since the increase in the degree of conjugation reduces the energies of RA’s LUMO, the driving force, i.e., the free energy change (-∆G_inj_) for electron injection, also decreases, because -∆G_inj_ is determined by the energy difference between the dye’s LUMO and the conduction band edge (CBE) of the semiconductor [107]. Therefore, -∆G_inj_ decreases from B to A to C.

According to the DFT calculations, it was determined that the photoinduced charge separation efficiency for complex B should be highest among the three complexes owing to the strongest mixing the LUMO of RA with the conduction band states of TiO_2_ and the highest -∆G_inj_. The amount of complex B can be increased by increasing the density of Ti-OH groups on the surface of TiO_2_, which can be achieved by hydroxylation treatment of the TiO_2_ surface. The EPR studies [99] showed that the photoinduced charge separation efficiency of RA on the TiO_2_ with the surface treatment was much higher than that on TiO_2_ without the treatment, strongly supporting the DFT calculations.

The above studies demonstrated that different anchoring modes for a carotenoid anchored on the surface of TiO_2_ significantly affect the conjugation of the carotenoid, which causes significant changes in the UV/vis absorption and the driving force for photo-induced ET. More importantly, the strengths of the coordination bonds formed between the dye and the Ti atom on the surface of TiO_2_ are significantly different for various anchoring modes, which affects the mixing of the dye’s excited state with conduction band states of TiO_2_ and, in turn, affects the efficiency of photo-induced ET. Since the λ_max_ of carotenoids are different for various anchoring modes, the colors of the carotenoids in cosmetics can be adjusted by choosing proper anchoring modes.

## 6. Conclusions

For effective application of carotenoids in the pharmaceutic, food and cosmetic industries, supramolecular carotenoid complexes are prepared to overcome some serious shortcomings such as low bioavailability and instability to light, high temperature, oxygen and metal ions. The antioxidant activities of the complexes are very important from the point of view of their application and storage stability. Since a large amount of supplement of carotenoids may cause side effects, a small amount of supplement of carotenoids encapsulated in delivery systems with increased antioxidant activity and reduced costs is desired.

The antioxidant activity of a carotenoid is exponentially dependent on the oxidation potential of the carotenoid. The oxidation potential of a carotenoid is related to the conjugation length and donor/acceptor substituents of the carotenoid. Carotenoids with shorter conjugation lengths and containing electron-withdrawing groups have higher oxidation potentials. This conclusion is important for the selection or synthesis of carotenoids in the pharmaceutical, food and cosmetic industries.

The polarity of the environment for a carotenoid in a supramolecular carotenoid complex also significantly affects the oxidation potential. The oxidation potential in a non-polar environment is much higher than in a polar environment. The difference in the oxidation potential of a carotenoid in a non-polar solvent with ε of about 2 versus that in a very polar solvent with ε of about 80, can be as large as 0.6 V. These results are independent of the symmetries and chain lengths of the studied carotenoids, and are basically applicable to all drugs, nutrients or cosmetics containing conjugated systems. However, the position and orientation of a carotenoid in a complex are to be considered in the determination of the radical scavenging ability as some positions or orientations may block the access of the radical to the most acidic protons of the carotenoid.

The chemical bonds or anchoring modes of carotenoids in drug delivery systems significantly affect the photostability and absorption of the carotenoids. The photoinduced charge separation efficiency of a carotenoid H-bonded to a host depends on whether the carotenoid acts as the H-bond donor or acceptor. The efficiency is much higher for the carotenoid acting as the H-bond donor than acceptor. For a carotenoid physisorbed on the surface of a host, the efficiency can be very low, especially if the environment is nonpolar, which results in much higher photostability. The different anchoring modes of a carotenoid on the surface of a semiconductor affect the degree of conjugation of the carotenoid, the driving force for ET and the mixing between the carotenoid’s LUMO and the conduction bands of the semiconductor, which in turn affect the photoinduced charge separation efficiency and photostability of the carotenoid. The maximum absorption wavelength of a carotenoid is also different for various anchoring modes, which can be exploited to adjust the color of the carotenoid in a product.

As a conclusion, this paper summarizes the most recent progress and useful information for fellow researchers, so as to enhance the utilization of carotenoids in the pharmaceutical, food and cosmetic industries. It is recommended that considerable research be required on the supramolecular carotenoid complexes, which can deliver more confidence in their utilization as drugs, nutrients and cosmetics.

## Figures and Tables

**Figure 1 antioxidants-09-00625-f001:**
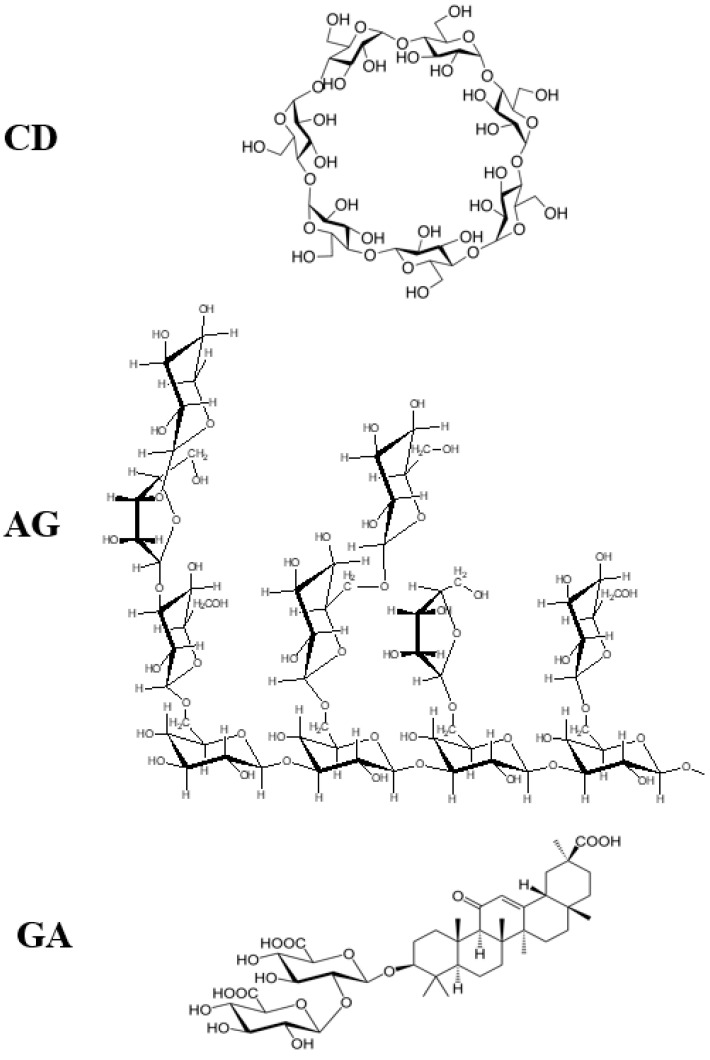
Structures of cyclodextrin (CD), arabinogalactan (AG) and β-glycyrrhizic acid (GA).

**Figure 2 antioxidants-09-00625-f002:**
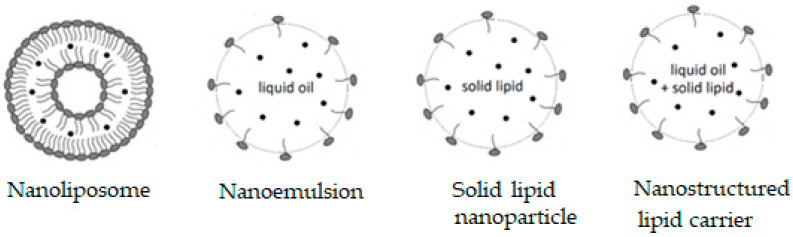
Lipid-based nanocarriers.

**Figure 3 antioxidants-09-00625-f003:**
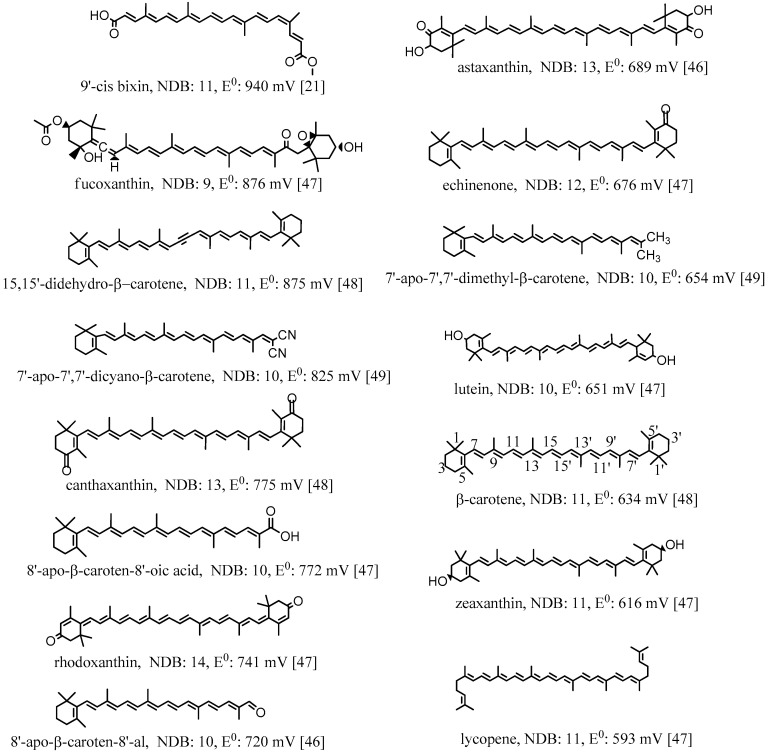
Structures, number of double bonds (NDB) and first oxidation potentials (vs. SCE) of selected carotenoids. The oxidation potentials were measured in CH_2_Cl_2_ by the method of cyclic voltammetry (CV) with measurement error ±10 mV. The reference electrode used in the measurements was saturated calomel electrode (SCE). Calibration with ferrocene gave the potentials corrected to SCE [46,48]. 86 mV was added to the potentials reported in Ref. [47] to correct for the absence of ferrocene calibration. This addition was shown to be needed in the examples given in Ref. [46]. The carbon atoms are numbered for β-carotene.

**Figure 4 antioxidants-09-00625-f004:**
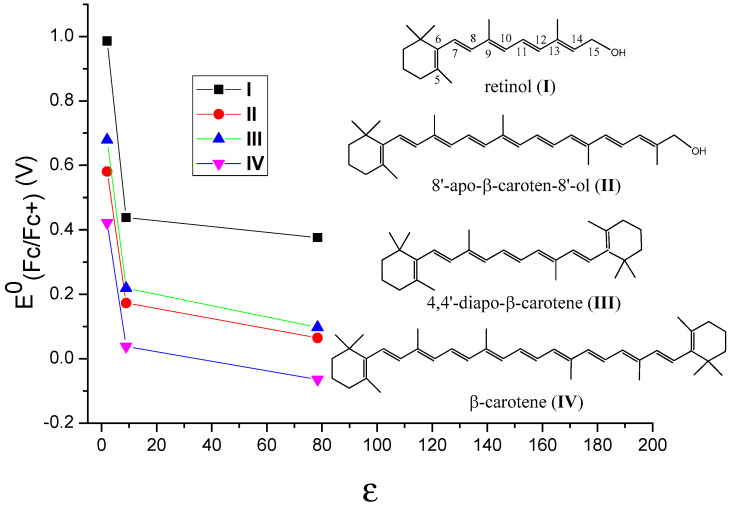
The change of the calculated oxidation potentials vs Fc/Fc+ (E^0^(Fc/Fc^+^)) with the dielectric constants (ε) of the solvents for retinol (I), 8’-apo-β-caroten-8’-ol (II), 4,4’-diapo-β-carotene (III) and β-carotene (IV). Black for retinol, red for 4,4’-diapo-β-carotene, green for 8’-apo-β-caroten-8’-ol and blue for β-carotene. Adapted from Ref. [51].

**Figure 5 antioxidants-09-00625-f005:**
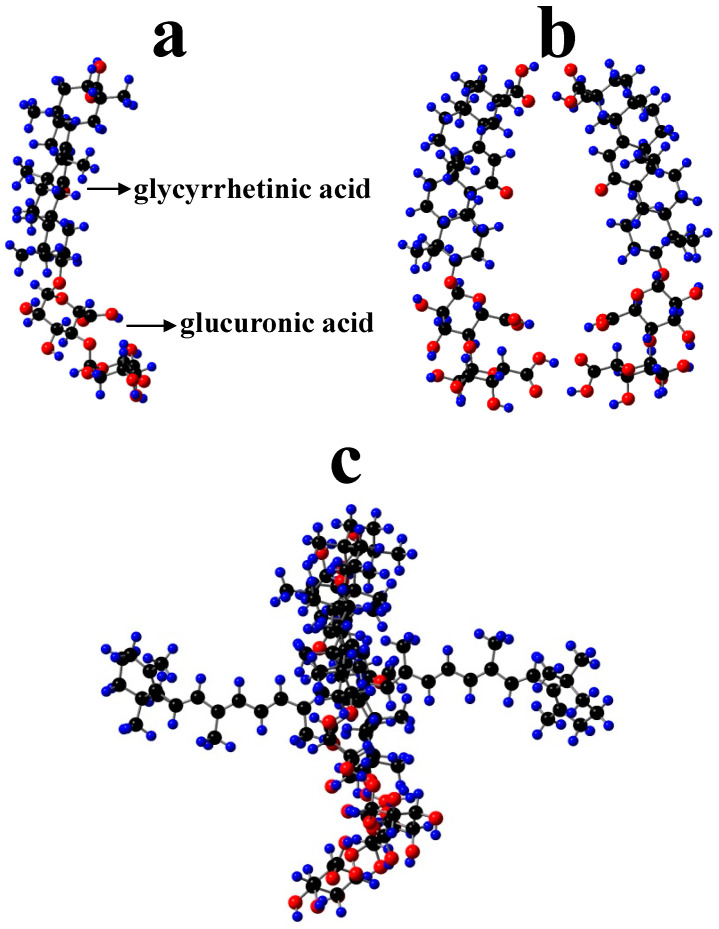
Optimized structures of (**a**) GA (β-glycyrrhizic acid) showing glycyrrhetinic acid residue (nonpolar) and glucuronic acid residues (polar), (**b**) dimer of GA and (**c**) β-carotene-GA complex by AM1 semi-empirical method. H, blue; O, red; C, black.

**Figure 6 antioxidants-09-00625-f006:**
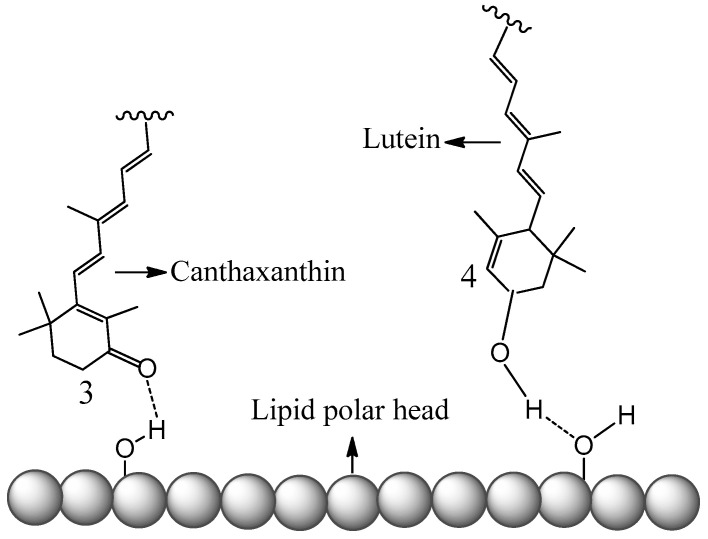
Schematic representation of canthaxanthin and lutein H-bonded to the surface of the lipid polar head.

**Figure 7 antioxidants-09-00625-f007:**
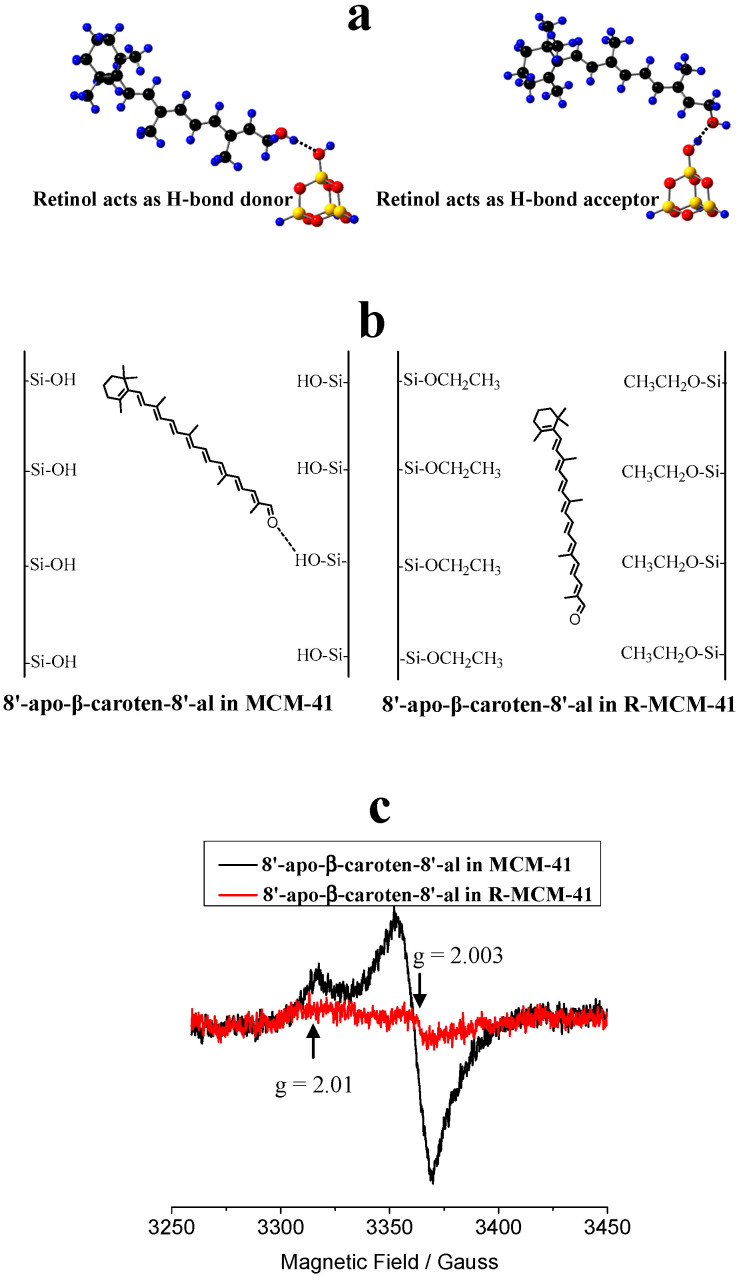
To increase the photostability of the carotenoids in supramolecular complexes, the formation of H-bonds should be avoided, especially when the carotenoid acts as a H-bond donor. Nonpolar environments also favor photostability. (**a**) Two types of H-bonds for retinol adsorbed on the surface of MCM-41. H: blue, O: red, C: black and Si: yellow. Adapted from Ref. [73]. (**b**) Cartoons for 8’-apo-β-caroten-8’-al imbedded in the pores of MCM-41 and in the modified surface of MCM-41 with –SiOH groups replaced by –SiOCH_2_CH_3_ groups designated as R-MCM-41. (**c**) EPR spectra after 10 min photo-irradiation of 8’-apo-β-caroten-8’-al adsorbed on activated MCM-41 and R-MCM-41, measured at 77 K. The backgrounds before photo-irradiation are subtracted. Adapted from Ref. [73]. Explanation of result according to [73]: the strong peak with g value of about 2.003 and peak to peak width of about 15 Gauss is the typical signal of a carotenoid radical cation. The weak peak with g value of about 2.01 could be attributed to a peroxy radical produced in a reaction of the carotenoid radical cation with O_2_ (O_2_ cannot be removed completely even under high vacuum) during photo-irradiation, and/or due to the reaction of a neutral radical (the radical cation of a carotenoid can lose a proton under photo-irradiation to produce the neutral radical) with O_2_.

**Figure 8 antioxidants-09-00625-f008:**
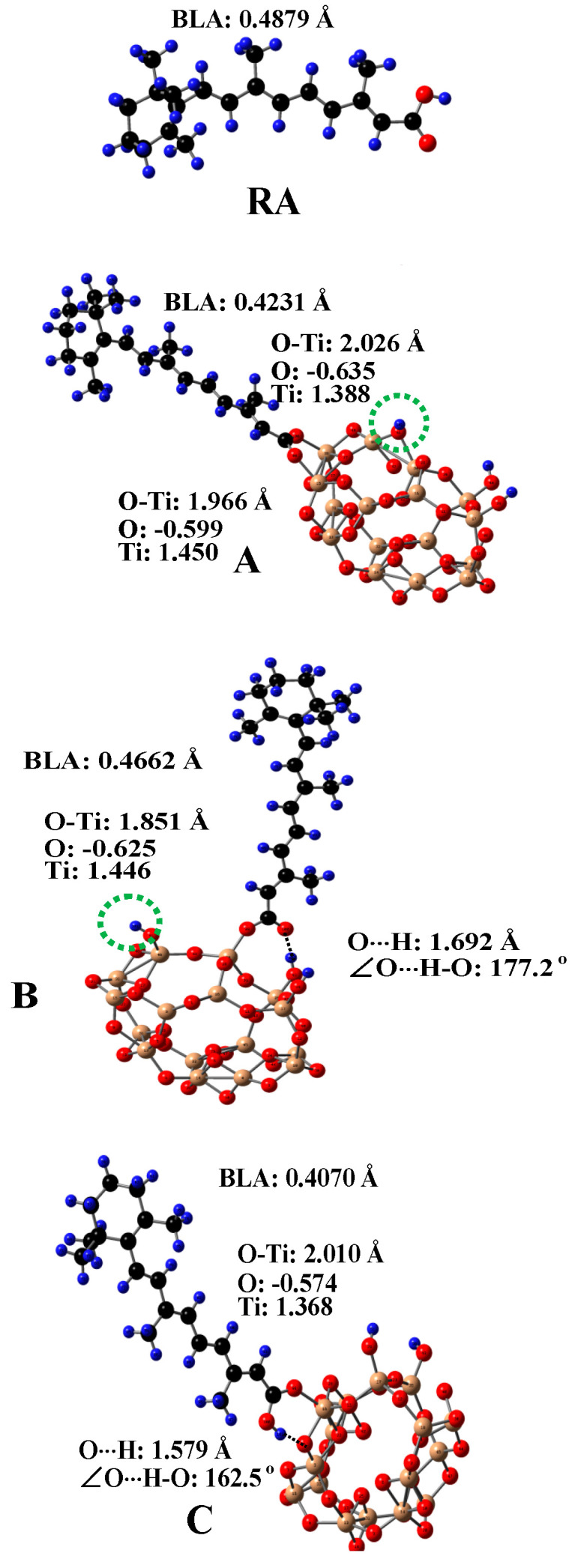
The optimized structures of retinoic acid (RA) and three complexes A, B, C for RA on the surface of a (TiO_2_)_16_ cluster. The bond lengths of O⋅⋅⋅H and O-Ti and ∠O⋅⋅⋅H-O for the complexes are shown in the figure. The calculated Mulliken charges on O and Ti for the O-Ti bonds are listed in the figure. The BLA (bond length alternation) value of RA for each complex is marked on the figure. The BLA value is an indication of the degree of conjugation in a molecule. The value decreases as the degree of conjugation increases. For A and B, the carboxylic acid group of RA loses a proton. The green dotted circle indicates the position of the lost proton. H, blue; O, red; C, black; Ti, gold. Adapted from Ref. [99].

**Table 1 antioxidants-09-00625-t001:** Calculated E^0^(Fc/Fc^+^) (V) values of the carotenoids in C_6_H_12_, CH_2_Cl_2_ and H_2_O.

	Car	I	II	III	IV
Solvent					
C_6_H_12_		0.986	0.581	0.679	0.422
CH_2_Cl_2_		0.438	0.173	0.219	0.0376
H_2_O		0.376	0.0645	0.0975	−0.0647

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
