# Peer review of "Antioxidant Activity in Supramolecular Carotenoid Complexes Favored by Nonpolar Environment and Disfavored by Hydrogen Bonding"

_antioxidants, 2020, doi:10.3390/antiox9070625_

Round 1
Reviewer 1 Report
The manuscript is a well written overview of the work of these researchers on the antioxidant capacity of carotenoids in a range of environments and will prove useful for a wide range of readers. It is structured very well, breaking down how conjugation length and substituents affect oxidation potentials, as well as discussing the effects of the polarity of the environment and also looking at carotenoids imbedded in molecular sieves and complexed with titanium dioxide and I definitely recommend it for publication in Antioxidants.
I have a few comments and very minor points, see list below.
- The page numbering on my copy had gone wrong with all pages saying ‘3 of 4’.
- I think it would be useful for the reader to review more of the literature in the introduction and perhaps in section 2, where more work has been published by other authors. For example, the oxidation potentials have been measured in aqueous micellar solutions by the Truscott group which, I believe was the first measurement in an aqueous environment. There has also been much work in membranes by the Skibsted group and a lot of older work comparing function with structure (i.e conjugation length and substituents) – see for example, Woodall et al BBA 1336, 33 (1997); Miller et al. FEBS Lett. 384, 240 (1996) and Mortensen & Skibsted J. Agric. Food Chem. 45, 2970 (1997).
- p. 7 Figure 4: I think the reference quoted for the figure should be 42 not 62.
- p. 13 Figure 7: Is reference 93 correct in the figure legend? Should it be 73?
- p. 16 Figure 8: O---H and <O---H-O have printed incorrectly in the legend.
- p. 17 Figure 9: The reference is given as 119 ad there are only 99 references.
- p. 17 Lines 480 & 484: TiO2 is bold.
- p. 17 Line 495: Remove ‘s’ from ‘carotenoids’.
Author Response
Response to Reviewer 1
Thank you very much for taking time to review our manuscript! Your comments have improved it, and your effort is greatly appreciated.
See our responses in blue in the attached file.

Reviewer 2 Report
In this manuscript, the authors review important evidences on the oxidation potentials of carotenoids and the different factors affecting them such as polar and non-polar environments.
Technically, the article is well written and coherent. Although Figure 2 and Figures 5-8 should be improved (quality, style of presentation, align, similar letters). From a scientific point of view, I have a few comments and suggestions:
- Authors should clarify better the significance of formation of H-bonds and consequently titania results. Few sentences or reorganization in the introduction will enhance these conclusions.
- Part 3: A table which summarizes oxidation potentials of different carotenoids and conditions can be helpful.
- Part 3: Authors talk about interaction between carotenoids and hydroperoxyl radicals. Any indication about scavenging rates of specific carotenoids toward peroxyl radicals?
- Line 219: The study [42] showed that…. , line 232: More importantly, the study [42] …., line 245: This recent study [42] also explains… Please replace/rewrite these sentences and other paragraphs in the manuscript thus beginning.
- Part 4: The author can provide a characteristic example (figure or data) with EPR results highlighting characteristic patterns.
Author Response
Response to Reviewer 2
Thank you very much for taking time to review our manuscript! Your comments have improved it, and your effort is greatly appreciated.
See our responses in blue in the attached file.
